# The Effects of Incorporating Ag-Zn Zeolite on the Surface Roughness and Hardness of Heat and Cold Cure Acrylic Resins

**Ali M. Aljafery [1], Ola M. Al-Jubouri [1], Zena J. Wally [1,\*], Rajaa M. Almusawi [1], Noor H. Abdulrudha [2] and Julfikar Haider [3,\*]**

[1] Department of Prosthodontic, Faculty of Dentistry, University of Kufa, Najaf 540011, Iraq; alim.aljaafri@uokufa.edu.iq (A.M.A.); olam.aljubouri@uokufa.edu.iq (O.M.A.-J.); rajaa.almusawi@uokufa.edu.iq (R.M.A.)

[2] Department of Basic Science, Faculty of Dentistry, University of Kufa, Najaf 540011, Iraq; noor.h.abd77@gmail.com

[3] Department of Engineering, Manchester Metropolitan University, Manchester M1 5GD, UK

[\*] Correspondence: zinah.alnuaimi@uokufa.edu.iq (Z.J.W.); J.Haider@mmu.ac.uk (J.H.); Tel.: +964-7718950063 (Z.J.W.); +44-1612473804 (J.H.)

**Abstract:** One of the most widely used materials for the fabrication of prosthetic dental parts is acrylic resin. Its reasonable mechanical and physical properties make it a popular material for a wide range of dental applications. Recently, many attempts have been made to improve the mechanical and biological properties of this material, such as by adding fibres, nanoparticles, and nanotubes. The current study aimed to evaluate the effects of adding an antimicrobial agent, Ag-Zn zeolite, on the surface roughness and hardness of the denture base resins. Ag-Zn zeolite particles were chemically prepared and added at different concentrations (0.50 wt.% and 0.75 wt.%) to the heat cure (HC) and cold cure (CC) acrylic resins. Zeolite particles were characterized and confirmed using X-ray diffraction (XRD) and Energy-Dispersive X-ray Spectroscopy (EDX) attached with a Scanning Electron Microscope (SEM). Sixty disk shape specimens (40 mm diameter and 2 mm thickness) were fabricated from the HC and CC resins with and without the zeolite. All the specimens were divided into two main groups based on the acrylic resins, then each was subdivided into three groups ($n = 10$) according to the concentration of the Ag-Zn zeolite. A surface roughness and a hardness tester were used to measure the surface finish and hardness of the specimens. The analysed data showed that the surface roughness values significantly decreased when 0.50 wt.% and 0.75 wt.% zeolite were incorporated in the HC resin specimens compared to the control group. However, this reduction was not significant in the case of CC resin, while the surface hardness was significantly improved after incorporating 0.50 wt.% and 0.75 wt.% zeolite for both the CC and HC resins. Incorporating Ag-Zn zeolite with acrylic resin materials could be beneficial for improving their surface finish and resistance to surface damage as defined by the higher hardness.

**Keywords:** heat cure acrylic resin; cold cure acrylic resin; Ag-Zn zeolite; denture base; surface roughness; surface hardness

## 1. Introduction

Acrylic resins are the preferred materials for several prosthodontic applications such as denture bases due to their ease of fabrication, low cost, low density, acceptable aesthetics, reasonable strength and suitability for the oral environment [1]. However, the mechanical properties of the acrylic resins still need to be improved where the removable denture bases are subjected to recurrent chewing forces in the oral cavity [2]. A surface roughness higher than the acceptable limit and degradation of the acrylic resins may also contribute to the adhesion and accumulation of microorganisms, which can increase the occurrence of denture-associated stomatitis [3]. Different trials have been reported to improve the strength properties of acrylic resins by integrating them with fibres, nanoparticles and

nanotubes [1,4]. Nevertheless, researchers are still seeking biocompatible additive materials to improve the mechanical and biological properties of the acrylic resins.

The deposition of biofilms on the surface of the acrylic denture bases is enhanced by the properties of the material, especially its porosity, irregularity and absorbency. *Candida albicans* yeast is the most common contributor for fungal infection affecting the oral mucosal tissues, especially in elderly patients fitted with complete dentures [2]. Although antifungal treatments either systemic or topical have been suggested to reduce the fungal infection, many barriers may limit their use such as microbial resistance, poor oral hygiene, cost and the strict routine of local treatment application with proper denture cleaning [5].

Incorporating different biomaterials with the acrylic resin has been reviewed in the literature, such as titanium dioxide [6] and zeolite [7]. Using zeolite as an antimicrobial agent with acrylic resins has shown the potential to elute agents from resins, eliminating bacterial, fungal and yeast infections [2,7]. Zeolite has also been shown to reduce the bacterial growth when combined with nonacrylic materials such as dental root fillings [8] and soft liners [9]. It is also used as a coating on titanium alloy implants and has been shown to diminish the incidence of peri-implant infections [10].

Zeolite is a naturally occurring mineral with unique properties, including low toxicity and lack of odour or flavour; therefore, it is used safely as a dietary supplement and in various medical treatments [2,5]. Zeolites are hydrated microporous aluminosilicates with an open framework structure and a negatively charged surface consisting of tetrahedra made of silicon and aluminium, which are connected by sharing oxygen [11,12]. The negative charge in the zeolites is neutralised by metal cations, either alkali or alkaline earth such as sodium, potassium and calcium, accommodated in the pores and cavities of the zeolite. These cations are easy to exchange with others and strongly affect the structure of the zeolite and thus their properties [13,14]. Silver and zinc cations can also be deposited within the pores and cavities of the zeolite and, over time, the free cations can be replaced by other cations from its environment [2,7]. Zeolites can be Linde type A (LTA) or NaA or 4A types. A well-known synthesised zeolite having a pore size of 0.4 nm, Si/Al molar ratio of 1 and high exchange capacity to other metal cations, 4A zeolite has sodium ions in its structure, which is easily exchangeable with other cations such as potassium and calcium and can thus be used for various industrial and chemical applications [15]. Silver ion exchange zeolites have shown excellent antibacterial activity with polymers. The mechanism of the antimicrobial effect of zeolite is explained by the ionic exchange reaction of the antimicrobial cations within which the pores of the zeolite are located. The antimicrobial behaviour of Ag-Zn zeolites has attracted researchers to mix it with the acrylic resins [5].

Acrylic resins currently do not contain any antimicrobial properties. By adding antimicrobial agents, bacterial and fungal contamination can be reduced without damaging the strength properties of dental materials [5]. The influence of adding different types of zeolite fillers on some mechanical properties of polymers has also been studied. Zeolite modified by 4-(dimethylamino) benzene diazonium cations was found to improve the compressive and flexural strength of the resin material [16]. The addition of 13X zeolite into heat cure acrylic resin showed no impact on the impact strength, transverse strength, surface hardness, surface roughness or colour [5]. In another study, silver-zinc zeolite decreased the flexural strength [17], and tensile and bending strength of heat-polymerized acrylic resin [18]. The application of zeolite in the medical fields has been reviewed in recent articles [19,20]. Moreover, the addition of silver nanoparticles to acrylic resin decreased the flexural strength of cured material, and this reduction was influenced by the concentration of nanoparticles [21]. Nanostructured silver vanadate ($\beta$-AgV03) also improved the surface hardness and increased the compressive strength of acrylic resin [22].

Although adding zeolite to the acrylic resin materials showed effective microbial inhibition, there is no clear evidence of its impact on the mechanical or surface properties of the acrylic resins, and it demands further investigation for dental applications. Furthermore,

no studies on the surface characteristics of 4A zeolite combined with heat cure (HC) and cold cure (CC) acrylic resins were found.

It is important to investigate the effect of the addition of zeolite as an antimicrobial agent in different types of PMMA for dental prostheses which are placed under recurrent pressure inside the patient's mouth. Other than providing the antimicrobial characteristics by zeolite, it is also important to ensure that zeolite addition is not going to affect the mechanical properties such as hardness or surface properties such as surface roughness of the PMMA resin. Unfortunately, there is limited research concerning the addition of zeolite material to the denture base resins [2]. Thus, this study aimed to evaluate the effects of adding two different concentrations (0.50 wt.% and 0.75 wt.%) of the 4A zeolite on the surface and microstructural characteristics of the HC and CC acrylic resins. The null hypothesis was that adding 4A zeolite to the HC and CC resins will not change their surface roughness and hardness.

## 2. Materials and Methods

### 2.1. Preparation of Ag-Zn Form A Zeolite

An amount of 10 g of 4A zeolite was hydrothermally prepared at 100 °C with a gel formula of 4 $Na_2O$:1 $Al_2O_3$:2 $SiO_2$:240 $H_2O$ and a Si/Al molar ratio of 1. The used chemicals and full synthesis procedures can be found in detail in [13,15]. The 4A zeolite exchanged form with Ag-Zn was prepared by the ion-exchange process following the procedure mentioned in [5,12]. Amounts of 5 g of zinc acetate (Central Drug House, India) and 0.5 g of silver acetate (Honey well, Germany) were completely dissolved in 100 g of deionised water. Then, 10 g of 4A zeolite was added to the Ag-Zn acetate solution and kept under continuous mixing for 150 min. However, according to [12,15], the ion-exchange time can be in the range of 120–1440 min. After that, the zeolite exchanged form was filtered and dried at 110 °C. Then, the dried powder was collected for characterization.

### 2.2. Characterisation of Ag-Zn Form A Zeolite

EDX using a TESCAN VEGA3 LM Oxford instrument was used to characterise the prepared particles and to find the percentage of Ag and Zn up-take by the zeolite. SEM was also conducted using the same instrument for EDX after coating the sample with a thin gold layer. In addition, the prepared zeolite powder was characterised by an XRD-type Shimadzu SRD 6000 at a 2θ ranging from 5 to 50° with a scan speed of 3° $min^{-1}$. The average crystal size of the zeolite sample was measured using Icy_all software consisting of the ImageJ option based on the SEM images [14,23]. Fifty points were randomly selected to conduct the analysis of crystal size by the software.

### 2.3. Specimens Fabrication and Grouping

Disk shaped stainless-steel patterns with 40 mm diameter and 2 mm thickness were used as a mould to prepare specimens from hot (TERMOPOLIMERIZABLE, VERACRIL-New SteticS. A. Colombia) and cold (AUTOPOLIMERIZABLE, VERACRIL-New SteticS. A. Colombia) cure acrylic resin. All metal patterns were placed in the dental flasks after adding the first layer of dental stone (type 3 ELITE MODEL- zhermack, Italy). Following the stone setting, the patterns were then removed from the flasks. For the experimental groups, Ag-Zn zeolite was added to the acrylic powder in two concentrations (0.5 wt.% and 0.75 wt.%) for both the HC and CC acrylic resins. The concentrations were chosen based on the studies in the literature [5,24,25] as a high concentration can significantly reduce the mechanical strength. According to the manufacturing instructions, the HC and CC acrylic resins were mixed (Powder: Liquid ratio is 2:1), and the specimens were packed and flasked. A long curing cycle was used to cure the HC acrylic resin. The curing temperature was programmed at 100 °C for 8 h, while the cold cure acrylic resin was allowed to polymerise at room temperature. A control group (without Ag-Zn zeolite) for both the HC and CC acrylic resins was also prepared using the same fabrication method

applied in the experimental groups. After that, all specimens were stored in water for 24 h without finishing and polishing.

The overall specimen number was 60 and the specimens were divided into 6 groups (*n* = 10) according to the type of acrylic resin and the concentration of Ag-Zn zeolite. Table 1 presents details of the material groups prepared for this study.

**Table 1.** Material groups used in the study.

| Group ID | Group Name | Acrylic Resin | Ag-Zn Zeolite (wt.%) |
|----------|------------|---------------|----------------------|
| A | HC Control group | Heat cure | 0.00 |
| B | HC0.50 group | Heat cure | 0.50 |
| C | HC0.75 group | Heat cure | 0.75 |
| D | CC Control group | Cold cure | 0.00 |
| E | CC0.50 group | Cold cure | 0.50 |
| F | CC0.75 group | Cold cure | 0.75 |

### 2.4. Surface Roughness Testing Procedure

A roughness tester (HSR 210, Hensgrand, Jinan, China) was used to evaluate the surface roughness of the specimens on a length of 2 mm. This test was performed on ten specimens for both HC and CC acrylic resins with and without 0.50 wt.% and 0.75 wt.% zeolite. Prior to testing, samples were immersed in distilled water at 37 °C for 48 h until testing [5]. To accurately measure the microgeometry of the surface, the specimens were placed on a stable bench during the test. The surface analyser of the testing machine scaled the surface irregularities of the specimens by recording all peaks and intervals to trace the profile of the surface irregularities. In each specimen, measurements were taken at four different locations and the mean of these readings was calculated.

### 2.5. Surface Hardness Testing Procedure

The hardness testing machine (shore D) was used to assess the hardness of the acrylic resins. It consisted of a blunt-pointed indenter of 0.8 mm in diameter that tapered to a diameter of 1.6 mm. The indenter was attached to a digital scale which was graduated from 0 to 100 units. The indenter was pressed down with substantial force and high speed. Measurements were taken directly from the digital scale reading. Four measurements were taken on different areas of each specimen (the same selected area of each specimen) and the average of the four readings was calculated [5].

### 2.6. Statistical Analysis

The collected data were displayed as mean $\pm$ standard deviation (SD). Mean values of the experimental groups were compared to the control group. Data were analysed by GraphPad Prism 9.2.0 software using one-way ANOVA with a Tukey's multiple comparison post-hoc test. $p < 0.05$ was determined to identify statistically significant differences.

## 3. Results
### 3.1. Characterisation of Ag-Zn Form A Zeolite

The XRD pattern of the prepared 4A zeolite is shown in Figure 1 within a 2θ range of 50°. The sharp and well-developed peaks within the range of 2θ suggested that a high degree of crystallinity and a highly ordered structure in the zeolite were formed. The peaks found at the 2θ values of 7°, 10.4°, 12.8°, 16.5°, 21.6°, 24°, 26.2°, 27°, 30°, 31°, 31.1°, 32.5°, 33.5° and 34.3° match well to the corresponding crystal planes of (200), (220), (222), (420), (440), (600), (622), (640), (642), (694), (660), (840), (842) and (664) mentioned in the literature [26].

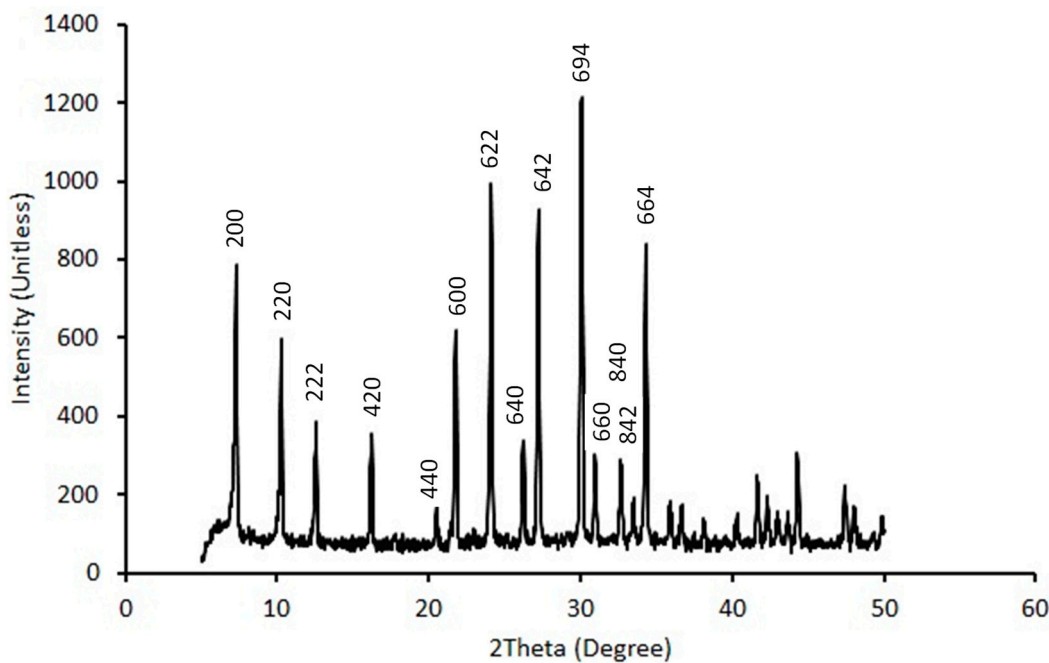

**Figure 1.** XRD pattern of the prepared 4A zeolite.

The crystallinity result was confirmed by the SEM images in Figure 2a, showing the morphology of the produced 4A zeolite as fully defined cubic crystals. Compared to the 4A zeolite, the SEM image of the Ag-Zn form A zeolite shown in Figure 2b revealed that the ion-exchange step made on the 4A zeolite did not noticeably change the morphology of the crystals. The average crystal sizes of the 4A zeolite and the Ag-Zn form A zeolite were 19.02 and 18.88 μm, respectively.

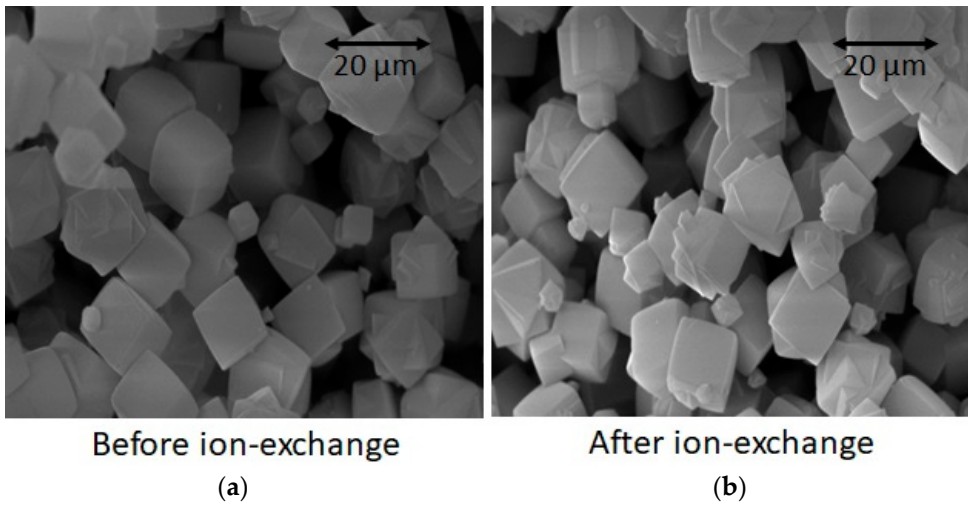

**Figure 2.** SEM images of (**a**) 4A zeolite and (**b**) Ag-Zn form A zeolite.

Table 2 and Figure 3 show the elemental analysis of the zeolite before and after the ion-exchange process, where the Si/Al ratio remained at 1. The EDX results showed that Ag-Zn form A zeolite contained Ag (4.34 wt.%) and Zn (8.43 wt.%). This confirmed the inclusion of Ag and Zn within the 4A zeolite. It was also noticed that weight percentages of the other elements in the 4A zeolite were slightly reduced.

**Table 2.** Elemental analysis of the prepared A zeolite before and after ion-exchange process.

| Elements | Ag-Zn Form A Zeolite (wt.%) | |
|---|---|---|
| | **Before Ion-Exchange** | **After Ion-Exchange** |
| O | 45.322 | 44.38 |
| Na | 12.196 | 5.06 |
| Al | 20.896 | 17.30 |
| Si | 21.483 | 20.50 |
| Ag | - | 4.34 |
| Zn | - | 8.43 |

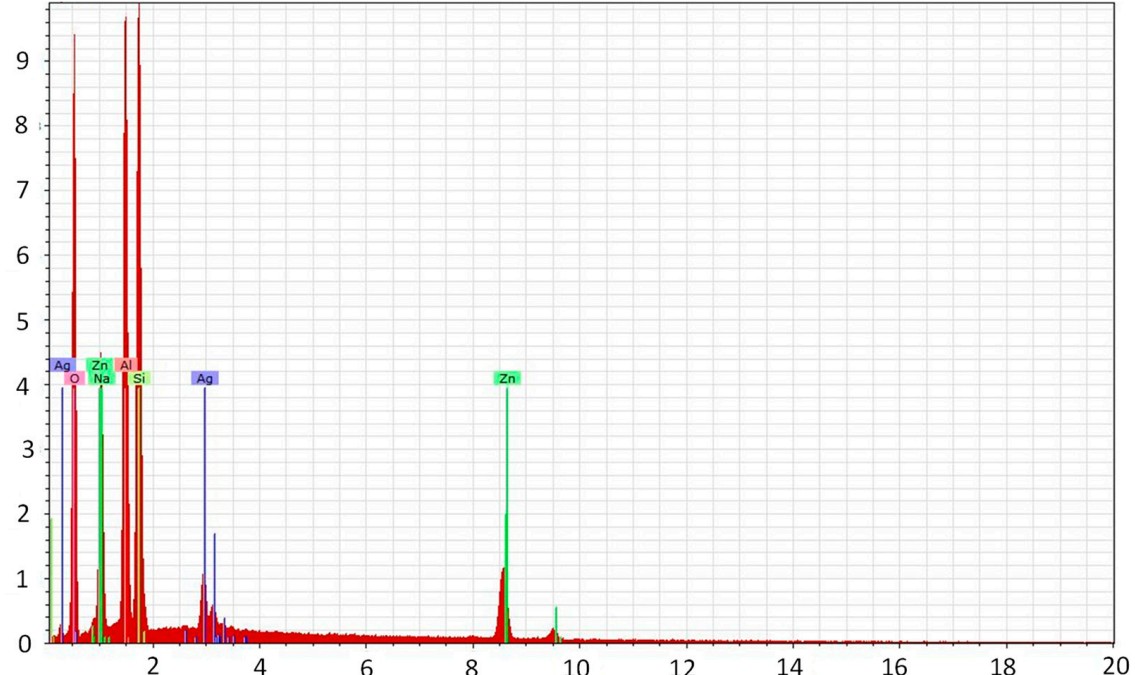

**Figure 3.** EDX results of Ag-Zn form A zeolite.

### 3.2. Surface Roughness

As shown in Table 3 and Figure 4, the incorporation of Ag-Zn zeolite in the HC acrylic resin at two different concentrations reduced the surface roughness values compared to the control group. The highest surface roughness value (0.7774 ± 0.0157) of the HC acrylic resin was found for the control group. In contrast, the lowest value (0.3578 ± 0.1925) was recorded for the HC resin incorporated with 0.50% Ag-Zn zeolite. There was a statistically significant difference between the surface roughness of the HC control group and the HC resin incorporated with both 0.50 wt.% and 0.75 wt.% Ag-Zn zeolites. However, the difference was nonsignificant between 0.50 wt.% and 0.75 wt.% Ag-Zn zeolite addition.

**Table 3.** Mean values of surface roughness in the HC acrylic resin incorporated with Ag-Zn zeolite at different percentages (0.50% and 0.75% $w/w$) and without any Ag-Zn zeolite (control group). Different superscript letters represent significant statistical differences ($p < 0.05$).

| Groups | HC Control | HC0.50 | HC0.75 |
|---|---|---|---|
| Number of data | 10 | 10 | 10 |
| Mean roughness (µm) | 0.7774 [a] | 0.3578 [b] | 0.5022 [b] |
| Std. Deviation (µm) | 0.0157 | 0.1925 | 0.1256 |
| Minimum (µm) | 0.752 | 0.202 | 0.328 |
| Maximum (µm) | 0.791 | 0.687 | 0.658 |

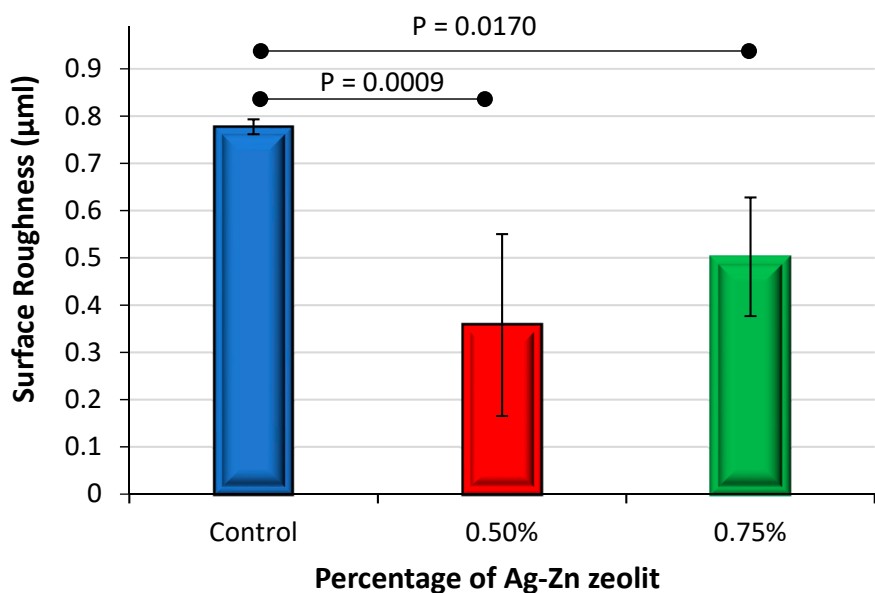

**Figure 4.** Bar chart showing mean values of the surface roughness in the HC acrylic resin incorporated with Ag-Zn zeolite at different concentrations (0.50 wt.% and 0.75 wt.%) and without any Ag-Zn zeolite (control group). Horizontal lines connecting two bars show statistical significance ($p < 0.05$).

The incorporation of Ag-Zn zeolite in the CC resin at two different concentrations (0.50 wt.% and 0.75 wt.%) was also found to reduce the surface roughness value, as shown in Table 4 and Figure 5. However, this difference was not significant. The highest surface roughness value (0.8528 ± 0.1353) of the CC resin was reported for the control group, while the lowest value (0.7010 ± 0.0682) was recorded for the CC resin incorporated with 0.75 wt.% Ag-Zn zeolite.

**Table 4.** Mean values of surface roughness in the CC resin incorporated with Ag-Zn zeolite at different concentrations (0.50 wt.% and 0.75 wt.%) and without any Ag-Zn zeolite (control group). Different superscript letters represent statistically significant differences ($p < 0.05$).

| Groups | Control | 0.50% | 0.75% |
|---|---|---|---|
| Number of data | 10 | 10 | 10 |
| Mean roughness (μm) | 0.8528 [a] | 0.7916 [a] | 0.7010 [a] |
| Std. Deviation (μm) | 0.1353 | 0.0655 | 0.0682 |
| Minimum (μm) | 0.6440 | 0.7120 | 0.6090 |
| Maximum (μm) | 0.6440 | 0.8810 | 0.7950 |

As expected, the values of surface roughness in the CC acrylic resin were higher than those in the HC acrylic resin with and without with Ag-Zn zeolite incorporation, as shown in Figure 6.

The surfaces of the HC samples also looked smoother than those of the CC samples, as shown in Figure 7, in line with the measurements presented in Figure 6.

### 3.3. Surface Hardness

The incorporation of HC acrylic resin with Ag-Zn zeolite at two different concentrations significantly increased the surface hardness values compared to the control group, as shown in Table 5 and Figure 8. The lowest surface hardness value (74.30 ± 2.163) of the HC acrylic resin was recorded in the control group. In contrast, the highest value (82.78 ± 0.0157) was found with the HC resin incorporated with 0.50% Ag-Zn zeolite. Nevertheless, the difference was significant between 0.50 wt.% and 0.75 wt.% Ag-Zn zeolite addition.

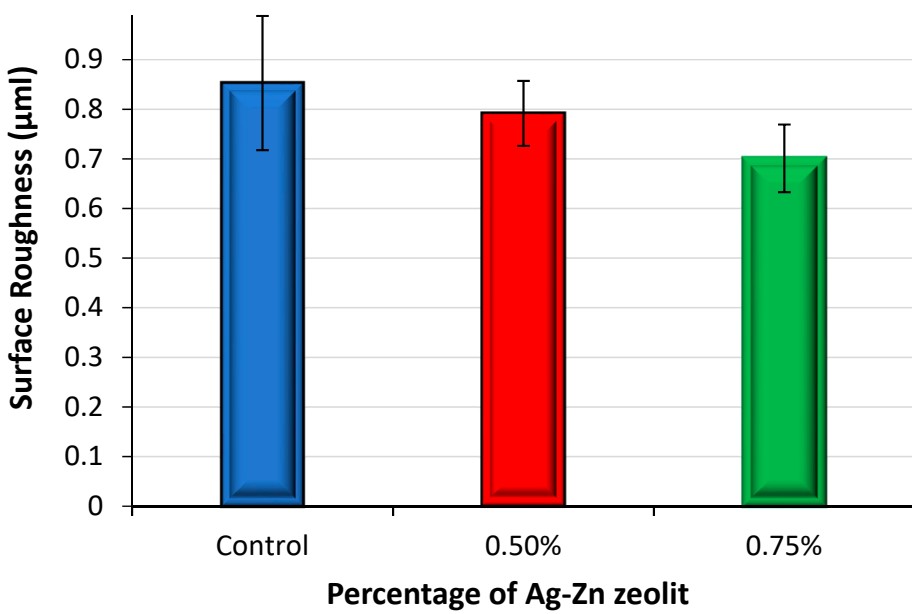

**Figure 5.** Bar chart showing mean values of surface roughness in the CC acrylic resin incorporated with Ag-Zn zeolite at different concentrations (0.50 wt.% and 0.75 wt.%) and without any Ag-Zn zeolite (control group).

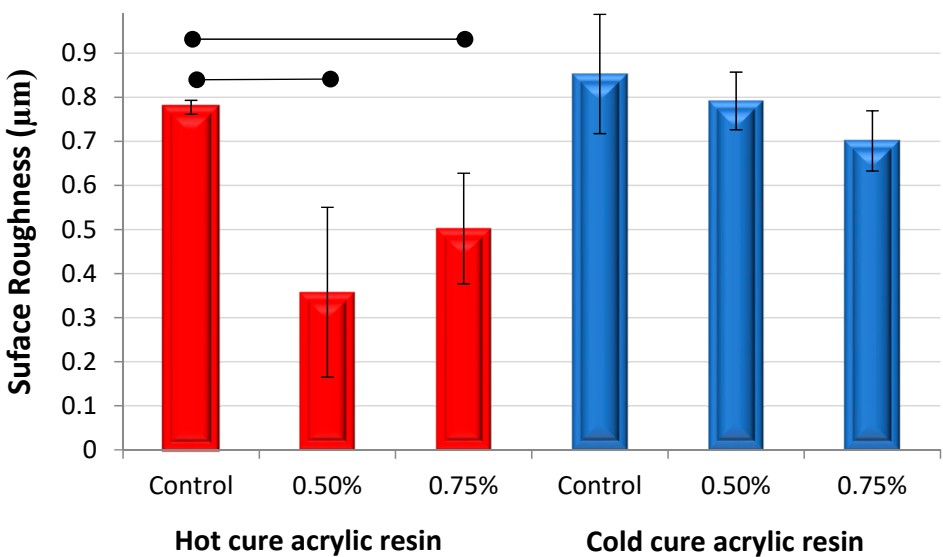

**Figure 6.** Bar chart showing the differences between the mean surface roughness values in the HC and CC acrylic resins incorporated with Ag-Zn zeolite at different concentrations (0.50 wt.% and 0.75 wt.%) and without any Ag-Zn zeolite (control group). Horizontal lines connecting two bars indicate statistical significance ($p < 0.05$).

Interestingly, the incorporation of CC resin with Ag-Zn zeolite at two different concentrations (0.50 wt.% and 0.75 wt.%) gradually improved the surface hardness value, as shown in Table 6 and Figure 9. The highest surface hardness value (76.18 ± 2.195) of the CC resin was reported for the 0.75 wt.% Ag-Zn zeolite, while the lowest value (70.17 ± 0.5049) was recorded for the control group.

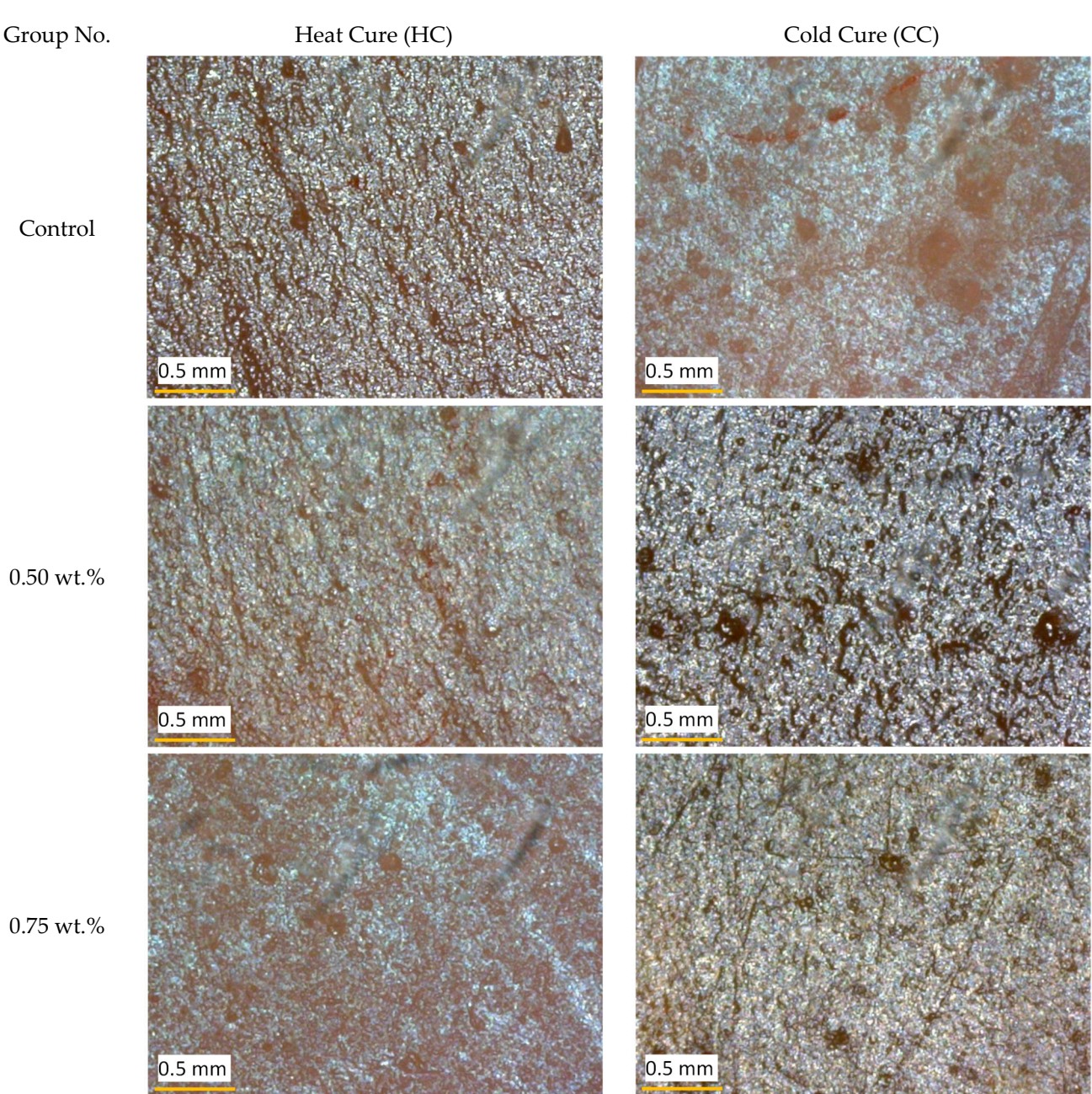

**Figure 7.** Optical micrographs (×2) of Heat Cure (HC) and Cold Cure (CC) samples' surfaces with and without Ag-Zn zeolite at different concentrations (0.50 wt.% and 0.75 wt.%).

**Table 5.** Mean values of surface hardness (Shore D) in the HC acrylic resin incorporated with Ag-Zn zeolite at different weight percentages (0.50 wt.% and 0.75 wt.%) and without any Ag-Zn zeolite (control group). Different superscript letters represent significant statistical differences ($p < 0.05$).

| Groups | HC Control | HC0.50 | HC0.75 |
|---|---|---|---|
| Number of data | 10 | 10 | 10 |
| Mean hardness (Shore D) | 74.30 [a] | 82.78 [b] | 79.20 [c] |
| Std. Deviation | 2.163 | 0.3701 | 0.7842 |
| Minimum | 71.20 | 82.30 | 78.00 |
| Maximum | 76.60 | 83.20 | 80.20 |

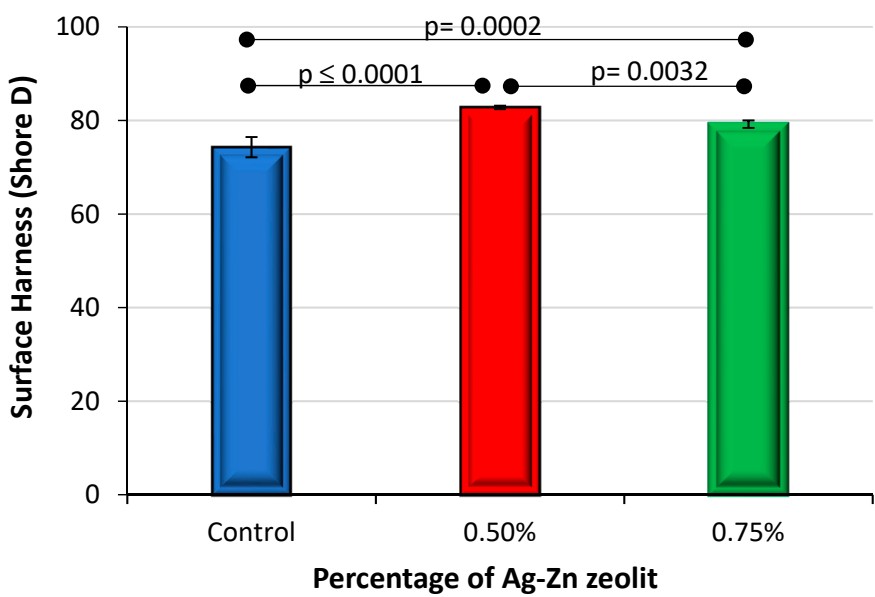

**Figure 8.** Bar chart showing mean values of the surface hardness in the HC acrylic resin incorporated with Ag-Zn zeolite at different concentrations (0.50 wt.% and 0.75 wt.%) and without any Ag-Zn zeolite (control group). Horizontal lines connecting two bars show statistical significance ($p < 0.05$).

**Table 6.** Mean values of surface hardness (Shore D) in the CC resin incorporated with Ag-Zn zeolite at different concentrations (0.50 wt.% and 0.75 wt.%) and without any Ag-Zn zeolite (control group). Different superscript letters represent significant statistical differences ($p < 0.05$).

| Groups | Control | 0.50% | 0.75% |
|---|---|---|---|
| Number of data | 10 | 10 | 10 |
| Mean hardness (Shore D) | 70.17 [a] | 75.48 [b] | 76.18 [b] |
| Std. Deviation | 0.5049 | 2.08 | 2.195 |
| Minimum | 69.69 | 72.10 | 73.50 |
| Maximum | 71.03 | 77.70 | 78.00 |

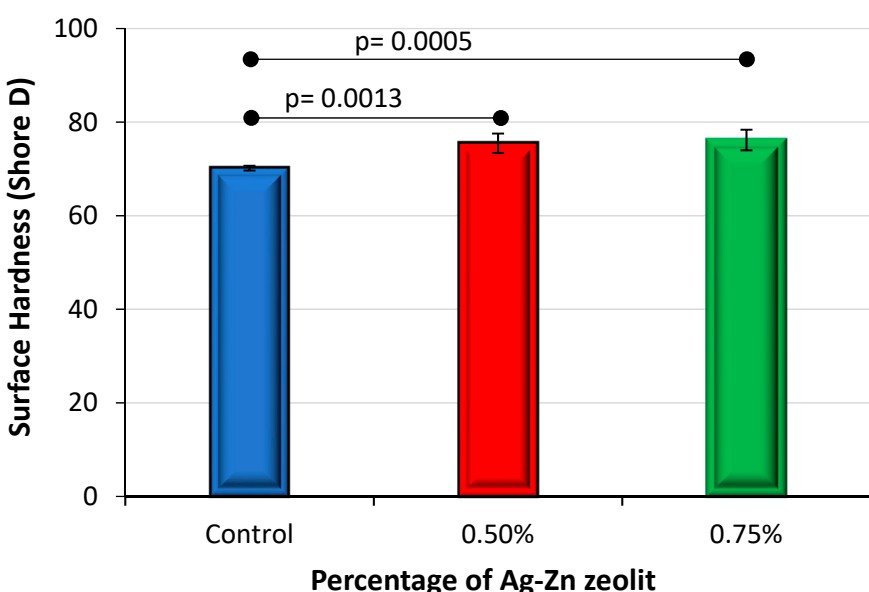

**Figure 9.** Bar chart showing mean values of surface hardness in the CC acrylic resin incorporated with Ag-Zn zeolite at different concentrations (0.50 wt.% and 0.75 wt.%) and without any Ag-Zn zeolite (control group). Horizontal lines connecting two bars show statistical significance ($p < 0.05$).

As expected, the surface hardness values of the HC acrylic resin were higher than those of the CC acrylic resin with and without incorporation of Ag-Zn zeolite, as shown in Figure 10.

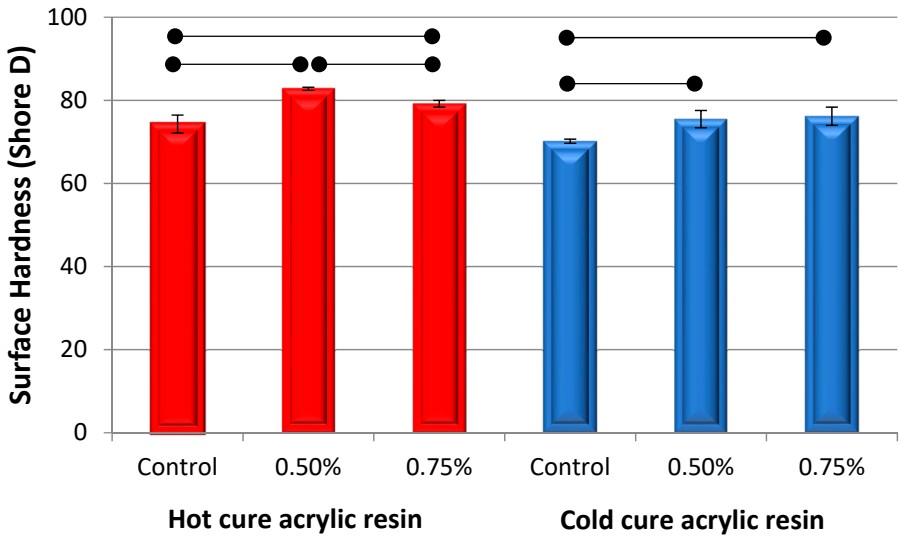

**Figure 10.** Bar chart showing the differences between the mean surface hardness values in the HC and CC acrylic resins incorporated with Ag-Zn zeolite at different concentrations (0.50 wt.% and 0.75 wt.%) and without any Ag-Zn zeolite (control group). Horizontal lines connecting two bars indicate statistical significance ($p < 0.05$).

## 4. Discussion

Materials that exhibit antimicrobial features have taken an increasing interest in the field of medicine and dentistry to prevent or reduce microorganism accumulation and subsequent infection. Antimicrobial zeolites have been incorporated with dental materials such as tissue conditioners, temporary fillings, and polymethyl methacrylate [2]. Incorporating zeolite with acrylic resin has effectively reduced bacterial and fungal infections in different studies [2,7].

Surface roughness plays a vital role in determining how a real object will interact with its environment. It was known to be a factor in the entrapment of microorganisms on acrylic surfaces. Higher numbers of microorganisms were observed on a rough surface than on a smooth surface. Irregularities and porosities present on the denture surface played a significant role in reducing the activity of denture-cleaning agents and hence an increased stain and plaque retention [27]. One of the main objectives for the resin restoration is that it produces a highly smooth surface without fine scratches to prevent microorganism collection on the external surface of the final restoration [28].

Although the incorporation of zeolite with acrylic resin materials showed effective antimicrobial behaviour, no other studies were found in the literature that investigated the effect of the zeolite on the surface roughness and hardness of both the HC and CC resin materials. Thus, to bring novelty, the present study aimed to evaluate the effect of incorporating two different percentages (0.50% and 0.75%) of 4A zeolite on the surface roughness and hardness of the acrylic resins. The zeolite was selected as a vehicle for the antimicrobial cations due to its properties, including prolonged antimicrobial activity, nontoxicity and lack of odour or flavour [29].

The results confirmed that the surface roughness values significantly decreased when 0.50 wt.% and 0.75 wt.% 4A zeolite were incorporated in the HC specimens compared to the corresponding control groups. However, this reduction was not significant in the case of CC acrylic resin. Therefore, the hypothesis is partially accepted.

The results of the current study showed a reduction in values of the surface roughness when the 4A zeolite was incorporated in both HC and CC acrylic specimens compared

to the control group. A significant difference between the control group and HC0.50 and HC0.75 groups was observed, which might be attributed to the HC acrylic resin having fewer porosities and voids which were usually generated between polymer chains during the curing process and uniform dispersion of the zeolite particles within the HC resin matrix. These porosities might be filled by even a small percentage of the Ag-Zn zeolite (0.50 wt.%) with small particle size (1888–1908 nm). Therefore, only a few particles might have appeared on the surface of the specimens resulting in a smooth surface. On the other hand, a nonsignificant difference was found between HC0.50 and HC0.75 groups. The HC0.75 group was smoother than the control but slightly rougher than the HC0.50 group. These results revealed that a slight increase in the Ag-Zn zeolite concentration to the HC acrylic resin might fill the porosities present between polymer chains until saturated, and then any excess zeolite material might gather on the external surface of the acrylic resin.

This result is comparable with that found by Alnamel and Mudhaffer when a small percentage of silicon dioxide nano-filler was added to the acrylic resin [30], and by Abdul-hamed and Mohammed when alumina was added to the acrylic resin [31]. Both concluded that when a small percentage of filler materials was added with the acrylic resin, a certain amount of filler would be incorporated within the acrylic resin matrix; with the increase in the percentage of the filler materials, more particles would appear on the external surface of the acrylic resin, leading to an increase in the surface roughness. However, the results of this study disagree with Azeeza and Fatah, who found that incorporating 0.50% of Ag-Zn zeolite into HC resin had a nonsignificant effect on the surface roughness [5].

The incorporation of cold cure acrylic resin with Ag-Zn zeolite at both tested concentrations was found to decrease the value of surface roughness compared to the control group, though the difference was nonsignificant between the CC control group and all the CC experimental groups. This might be related to the fact that the cold cure acrylic resin might produce many porosities and voids due to the evaporation of the monomer during the curing process, which filled with the Ag-Zn zeolite material that led to the production of surfaces smoother than those of the control group. The results also showed a nonsignificant difference between the control group and the CC0.50 group and a nonsignificant difference between the CC0.50 and CC0.75 groups. These results might be related to the fact that the addition of 0.75 wt.% Ag-Zn zeolite filled most of the porosities and voids present within the CC resin matrix, producing a smoother surface than the control group and the CC0.50 group.

In general, the values of surface roughness in the HC resin were lower than those in the CC resin with and without the Ag-Zn zeolite incorporation. This result agrees with Ogle et al.'s study, which showed that heat-polymerised resin had the smoothest surface compared to the self-polymerised and light-polymerised acrylic resins [32].

There were not enough studies in relation to the addition of Ag-Zn zeolite to acrylic denture base materials to further compare and contrast with this study. However, the beneficial effect of adding Ag-Zn zeolite to other materials was evidenced in the literature. The result of this study is comparable with Ari et al. who found decreasing surface roughness and dimensional change of the casting after the addition of zeolite to phosphate-bonded investment due to its molecular sieve property [33]. Furthermore, Sadeq and Hummudi found a significant improvement in shear bond strength of the HC silicon soft liner after incorporation of 0.50 wt.% and 0.75 wt.% Ag-Zn zeolite [24]. However, in the case of CC resin, the same authors found that only 0.75 wt.% Ag-Zn zeolite can generate a surface roughness significantly lower than that of the pure CC resin [24]. Ning et al. found that the surface roughness of zeolite-polyamide thin-film nanocomposite membranes (TFN) increased by increasing the zeolite loading as a result of the zeolite incorporation, which was characterised by a greater nodular appearance on the surface of TFN membranes [34]. Al-Azawi and Al-Naqash found that the incorporation of 0.50% of Ag-Zn zeolite into silicon impression material did not affect the setting time [35].

Although the results obtained in the present study demonstrated the valuable effect of Ag-Zn zeolite in reducing the surface roughness of the HC and CC acrylic resins compared

to the pure resins, one limitation of this study was that the surface roughness after the regular polishing procedure was not assessed here. Further investigations could be carried out to study the influence of Ag-Zn zeolite on the surface roughness of the acrylic resins before and after the polishing procedure. Hence, the incorporation of acrylic resins with such biocompatible and inexpensive materials could be promising for improving the surface properties of acrylic resin, particularly the HC one. A further in-depth understanding about the surfaces could be gained with an SEM or an optical surface profilometer.

The incorporation of fillers in plastics may alter the resistance to abrasion, but the hardness of the plastic matrix remains unchanged [36]. In the present study, the addition of 0.50 wt.% and 0.75 wt.% Ag-Zn zeolite significantly increased the surface hardness of heat and cold cure acrylic resin compared to the control group. The improvement in the surface hardness was dose-dependent. Zeolite consists of silicon oxide with fine particle sizes characterized by a greater surface area, providing the interstitial adhesion with the polymer chains. The random distribution of zeolite particles within polymer chains also leads to an improvement in the surface hardness. The results of this study are in agreement with Azeez and Fatah [5] who found that the addition of 0.5% of antimicrobial sliver-zinc zeolite into heat cure acrylic increased the surface hardness. Alnamel and Mudhaffar found an increase in surface hardness of acrylic resin after the addition of silicon dioxide nano-fillers [30].

There was a statistically high significant difference between the CCControl group and CC0.5 group, a high significant difference between the CCControl and CC0.75 groups, and a nonsignificant difference between the CC0.50 and CC0.75 groups. The possible explanation of these results could be attributed to the fine particle size of the zeolite material acting as a filler, which is uniformly distributed within the polymer matrix and fills the porosities which are usually generated due to the evaporation of monomers of cold cure acrylic resin during the polymerization process.

The results of this study are supported by the findings of Yasser and Fatah who found that the addition of zirconium nanoparticles to the acrylic-based soft liner caused an increase in shear bond strength [37]. Sadeq and Hummudi found that the incorporation of 0.5% and 0.75% of Ag-Zn Zeolite into cold soft liner had a significant effect on the shear bond strength [38]. On the contrary, it has been stated that currently available acrylic resin materials do not have any antimicrobial properties. With the addition of antimicrobial agents, bacterial and fungal contamination can be reduced without impairing the mechanical and surface properties of the dental materials [39]. However, the difference between these studies with the current study was the type of resin material used, types of filler and the mechanical tests used [25].

Overall, the HC and CC resins with the Ag-Zn zeolite infusion can provide a smoother, harder and abrasion-resistant surface along with the bacteria-killing capability, which can improve clinical performance and patient's satisfaction.

## 5. Conclusions

In this study, Ag-Zn zeolite was synthesised and characterised. The impact of adding different concentrations (0.50 wt.%, 0.75 wt.%) of Ag-Zn zeolite to the HC and CC acrylic resins on their surface roughness was studied compared to the corresponding pure resins. The crystallinity of the prepared Ag-Zn zeolite particles was confirmed by XRD, and average crystal sizes of 1902 nm and 1888 nm were recorded for the HC and CC resins, respectively. In general, all the specimens with Ag-Zn zeolite showed a lower surface roughness than the pure HC and CC resins. However, this reduction in roughness was only significant for the HC resin. The addition of 0.5 wt.% and 0.75 wt.% Ag-Zn zeolite significantly increased the surface hardness of both resins.

**Author Contributions:** Conceptualization, A.M.A., O.M.A.-J., Z.J.W. and R.M.A.; methodology, O.M.A.-J., Z.J.W., R.M.A. and N.H.A.; validation, O.M.A.-J., Z.J.W., R.M.A. and J.H.; formal analysis, Z.J.W.; investigation, O.M.A.-J., Z.J.W., R.M.A., N.H.A. and J.H.; writing—original draft preparation, O.M.A.-J., Z.J.W., N.H.A. and R.M.A.; writing—review and editing, Z.J.W., R.M.A. and J.H.; visualization, Z.J.W., R.M.A. and J.H.; supervision, J.H. All authors have read and agreed to the published version of the manuscript.

**Funding:** This research did not receive any specific grant from funding agencies in the public, commercial, or not-for-profit sectors.

**Data Availability Statement:** The data that support the findings of this study are available from the corresponding author upon reasonable request.

**Acknowledgments:** The authors would like to acknowledge the support from the University of Kufa for providing the experimental facility. On behalf of all the authors in this manuscript, we would like to dedicate this paper to the fourth author, Rajaa M. Almusawi's father (Mahdi Almusawi), who sadly passed away due to COVID-19. We deeply appreciate Almusawi's commitment and courage in completing this paper during this difficult time.

**Conflicts of Interest:** The authors declare no conflict of interest.

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
