# Peer review of "The Effects of Incorporating Ag-Zn Zeolite on the Surface Roughness and Hardness of Heat and Cold Cure Acrylic Resins"

_jcs, doi:10.3390/jcs6030085_

Round 1
Reviewer 1 Report
The authors evaluated the effects of incorporating Ag-Zn zeolite on acrylic resin's surface roughness. After reviewing the manuscript, I think it can be considered to publish in the Journal of Composites Science. However, there are some minor points that authors should consider to upgrade the quality of their work:
- The background information of Ag-Zn zeolite and the reason to mix with resin should be more clearly introduced.
- Are there any changes in XRD of zeolite before and after ion exchange with Ag-Zn?
- Some minor points:
- The name of bacteria species (Candida albicans, Page 2, Line 48-50) must be italic. There are also some typing errors in the entire manuscript and should be corrected.
- Page 6, Line 192-193: “Table 2” -> “Table 3”

Author Response
Dear Colleague,
We would like to thank the Editor and reviewers for spending time reviewing the article and providing constructive feedback. Having read the comments and consulted with the co-authors we have now made the appropriate amendments. We hope that the revised version meets your approval.

Reviewer 2 Report
This manuscript is well written.
Only a few minor suggestions:
In the title, I suggest omitting the word "evaluating"
In the abstract, please discuss if the changes were significant (lines 27-28), also in the results and discussion part.
Please discuss also the limitations of your research, novelty and implications for practice (abstract, discussion part).
Please discuss more deeply in the Introduction part previous research results dealing with the influence of additives on the surface roughness of resins.
The Material and Methods section is well written except surface roughness testing procedure - please be more precise about describing of this test.
In line 148 you talk about "Each specimen was tested four times" in Table 3 Number of data = 10, please explain.
Table 3 header should be before the table.
Line 159: The statistical
analyses of the experimental values were compared to the control group. - - please reformulate, statistical analyses of both exp. and contr. data...
Abbreviations - please check the manuscript, some are explained repeatedly (XRD - abstract, line 119, etc.)
In the discussion section please add more discussion with other authors dealing with the same topic.
Please check the template for some minor issues like figures (font), multiple references, tables, references.
Author Response

(The authors gave the same response as above.)
